# Hyperbolic Graph Convolutional Neural Networks

Ines Chami[*‡]    Rex Ying[*†]    Christopher Ré[†]    Jure Leskovec[†]

[†]Department of Computer Science, Stanford University
[‡]Institute for Computational and Mathematical Engineering, Stanford University
{chami, rexying, chrismre, jure}@cs.stanford.edu

## Abstract

Graph convolutional neural networks (GCNs) embed nodes in a graph into Euclidean space, which has been shown to incur a large distortion when embedding real-world graphs with scale-free or hierarchical structure. Hyperbolic geometry offers an exciting alternative, as it enables embeddings with much smaller distortion. However, extending GCNs to hyperbolic geometry presents several unique challenges because it is not clear how to define neural network operations, such as feature transformation and aggregation, in hyperbolic space. Furthermore, since input features are often Euclidean, it is unclear how to transform the features into hyperbolic embeddings with the right amount of curvature. Here we propose Hyperbolic Graph Convolutional Neural Network (HGCN), the first inductive hyperbolic GCN that leverages both the expressiveness of GCNs and hyperbolic geometry to learn inductive node representations for hierarchical and scale-free graphs. We derive GCNs operations in the hyperboloid model of hyperbolic space and map Euclidean input features to embeddings in hyperbolic spaces with different trainable curvature at each layer. Experiments demonstrate that HGCN learns embeddings that preserve hierarchical structure, and leads to improved performance when compared to Euclidean analogs, even with very low dimensional embeddings: compared to state-of-the-art GCNs, HGCN achieves an error reduction of up to 63.1% in ROC AUC for link prediction and of up to 47.5% in F1 score for node classification, also improving state-of-the art on the Pubmed dataset.

## 1   Introduction

Graph Convolutional Neural Networks (GCNs) are state-of-the-art models for representation learning in graphs, where nodes of the graph are embedded into points in Euclidean space [15, 21, 41, 45]. However, many real-world graphs, such as protein interaction networks and social networks, often exhibit scale-free or hierarchical structure [7, 50] and Euclidean embeddings, used by existing GCNs, have a high distortion when embedding such graphs [6, 32]. In particular, scale-free graphs have tree-like structure and in such graphs the graph volume, defined as the number of nodes within some radius to a center node, grows exponentially as a function of radius. However, the volume of balls in Euclidean space only grows polynomially with respect to the radius, which leads to high distortion embeddings [34, 35], while in hyperbolic space, this volume grows exponentially.

Hyperbolic geometry offers an exciting alternative as it enables embeddings with much smaller distortion when embedding scale-free and hierarchical graphs. However, current hyperbolic embedding techniques only account for the graph structure and do not leverage rich node features. For instance, Poincaré embeddings [29] capture the hyperbolic properties of real graphs by learning shallow embeddings with hyperbolic distance metric and Riemannian optimization. Compared to

---

[*]Equal contribution

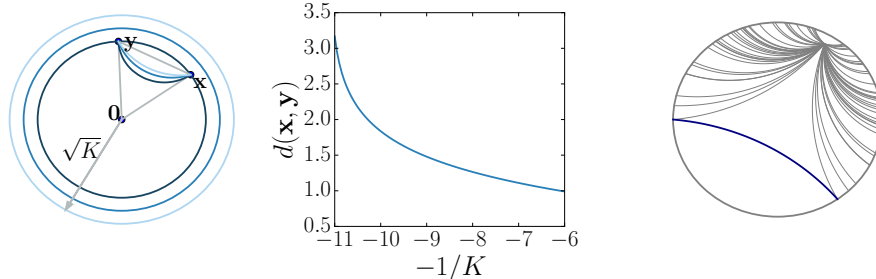

Figure 1: Left: Poincaré disk geodesics (shortest path) connecting **x** and **y** for different curvatures. As curvature $(-1/K)$ decreases, the distance between **x** and **y** increases, and the geodesics lines get closer to the origin. Center: Hyperbolic distance vs curvature. Right: Poincaré geodesic lines. x

deep alternatives such as GCNs, shallow embeddings do not take into account features of nodes, lack scalability, and lack inductive capability. Furthermore, in practice, optimization in hyperbolic space is challenging.

While extending GCNs to hyperbolic geometry has the potential to lead to more faithful embeddings and accurate models, it also poses many hard challenges: (1) Input node features are usually Euclidean, and it is not clear how to optimally use them as inputs to hyperbolic neural networks; (2) It is not clear how to perform set aggregation, a key step in message passing, in hyperbolic space; And (3) one needs to choose hyperbolic spaces with the right curvature at every layer of the GCN.

Here we solve the above challenges and propose *Hyperbolic Graph Convolutional Networks* (*HGCN*)[2], a class of graph representation learning models that combines the expressiveness of GCNs and hyperbolic geometry to learn improved representations for real-world hierarchical and scale-free graphs in inductive settings: (1) We derive the core operations of GCNs in the hyperboloid model of hyperbolic space to transform input features which lie in Euclidean space into hyperbolic embeddings; (2) We introduce a hyperbolic attention-based aggregation scheme that captures hierarchical structure of networks; (3) At different layers of HGCN we apply feature transformations in hyperbolic spaces of different trainable curvatures to learn low-distortion hyperbolic embeddings.

The transformation between different hyperbolic spaces at different layers allows HGCN to find the best geometry of hidden layers to achieve low distortion and high separation of class labels. Our approach jointly trains the weights for hyperbolic graph convolution operators, layer-wise curvatures and hyperbolic attention to learn inductive embeddings that reflect hierarchies in graphs.

Compared to Euclidean GCNs, HGCN offers improved expressiveness for hierarchical graph data. We demonstrate the efficacy of HGCN in link prediction and node classification tasks on a wide range of open graph datasets which exhibit different extent of hierarchical structure. Experiments show that HGCN significantly outperforms Euclidean-based state-of-the-art graph neural networks on scale-free graphs and reduces error from 11.5% up to 47.5% on node classification tasks and from 28.2% up to 63.1% on link prediction tasks. Furthermore, HGCN achieves new state-of-the-art results on the standard PUBMED benchmark. Finally, we analyze the notion of hierarchy learned by HGCN and show how the embedding geometry transforms from Euclidean features to hyperbolic embeddings.

## 2 Related Work

The problem of graph representation learning belongs to the field of geometric deep learning. There exist two major types of approaches: transductive shallow embeddings and inductive GCNs.

**Transductive, shallow embeddings**. The first type of approach attempts to optimize node embeddings as parameters by minimizing a reconstruction error. In other words, the mapping from nodes in a graph to embeddings is an embedding look-up. Examples include matrix factorization [3, 24] and random walk methods [12, 31]. Shallow embedding methods have also been developed in hyperbolic geometry [29, 30] for reconstructing trees [35] and graphs [5, 13, 22], or embedding text

**(Euclidean) Graph Neural Networks**. Instead of learning shallow embeddings, an alternative approach is to learn a mapping from input graph structure as well as node features to embeddings, parameterized by neural networks [15, 21, 25, 41, 45, 47]. While various Graph Neural Network architectures resolve the disadvantages of shallow embeddings, they generally embed nodes into a Euclidean space, which leads to a large distortion when embedding real-world graphs with scale-free or hierarchical structure. Our work builds on GNNs and extends them to hyperbolic geometry.

**Hyperbolic Neural Networks**. Hyperbolic geometry has been applied to neural networks, to problems of computer vision or natural language processing [8, 14, 18, 38]. More recently, hyperbolic neural networks [10] were proposed, where core neural network operations are in hyperbolic space. In contrast to previous work, we derive the core neural network operations in a more stable model of hyperbolic space, and propose new operations for set aggregation, which enables HGCN to perform graph convolutions with attention in hyperbolic space with trainable curvature. After NeurIPS 2019 announced accepted papers, we also became aware of the concurrently developed HGNN model [26] for learning GNNs in hyperbolic space. The main difference with our work is how our HGCN defines the architecture for neighborhood aggregation and uses a learnable curvature. Additionally, while [26] demonstrates strong performance on graph classification tasks and provides an elegant extension to dynamic graph embeddings, we focus on link prediction and node classification.

# 3 Background

**Problem setting**. Without loss of generality we describe graph representation learning on a single graph. Let $\mathcal{G} = (\mathcal{V}, \mathcal{E})$ be a graph with vertex set $\mathcal{V}$ and edge set $\mathcal{E}$, and let $(\mathbf{x}_i^{0,E})_{i \in \mathcal{V}}$ be $d$-dimensional input node features where $^0$ indicates the first layer. We use the superscript $^E$ to indicate that node features lie in a Euclidean space and use $^H$ to denote hyperbolic features. The goal in graph representation learning is to learn a mapping $f$ which maps nodes to embedding vectors:

$$f : (\mathcal{V}, \mathcal{E}, (\mathbf{x}_i^{0,E})_{i \in \mathcal{V}}) \to Z \in \mathbb{R}^{|\mathcal{V}| \times d'},$$

where $d' \ll |\mathcal{V}|$. These embeddings should capture both structural and semantic information and can then be used as input for downstream tasks such as node classification and link prediction.

**Graph Convolutional Neural Networks (GCNs)**. Let $\mathcal{N}(i) = \{j : (i, j) \in \mathcal{E}\}$ denote a set of neighbors of $i \in \mathcal{V}$, $(W^\ell, \mathbf{b}^\ell)$ be weights and bias parameters for layer $\ell$, and $\sigma(\cdot)$ be a non-linear activation function. General GCN message passing rule at layer $\ell$ for node $i$ then consists of:

$$\mathbf{h}_i^{\ell,E} = W^\ell \mathbf{x}_i^{\ell-1,E} + \mathbf{b}^\ell \qquad \text{(feature transform)} \qquad (1)$$

$$\mathbf{x}_i^{\ell,E} = \sigma(\mathbf{h}_i^{\ell,E} + \sum_{j \in \mathcal{N}(i)} w_{ij} \mathbf{h}_j^{\ell,E}) \qquad \text{(neighborhood aggregation)} \qquad (2)$$

where aggregation weights $w_{ij}$ can be computed using different mechanisms [15, 21, 41]. Message passing is then performed for multiple layers to propagate messages over network neighborhoods. Unlike shallow methods, GCNs leverage node features and can be applied to unseen nodes/graphs in inductive settings.

**The hyperboloid model of hyperbolic space**. We review basic concepts of hyperbolic geometry that serve as building blocks for HGCN. Hyperbolic geometry is a non-Euclidean geometry with a constant negative curvature, where curvature measures how a geometric object deviates from a flat plane (*cf.* [33] for an introduction to differential geometry). Here, we work with the hyperboloid model for its simplicity and its numerical stability [30]. We review results for any constant negative curvature, as this allows us to learn curvature as a model parameter, leading to more stable optimization (*cf.* Section 4.5 for more details).

**Hyperboloid manifold**. We first introduce our notation for the hyperboloid model of hyperbolic space. Let $\langle ., . \rangle_{\mathcal{L}} : \mathbb{R}^{d+1} \times \mathbb{R}^{d+1} \to \mathbb{R}$ denote the Minkowski inner product, $\langle \mathbf{x}, \mathbf{y} \rangle_{\mathcal{L}} := -x_0 y_0 +$

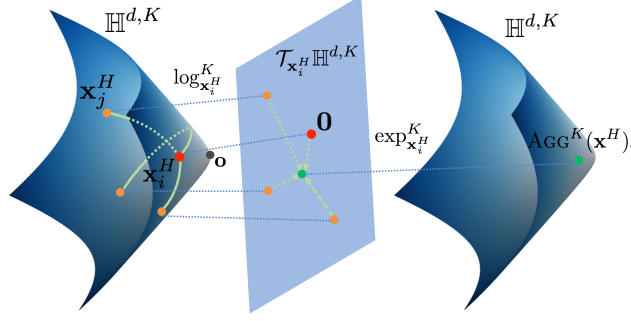

Figure 2: HGCN neighborhood aggregation (Eq. 9) first maps messages/embeddings to the tangent space, performs the aggregation in the tangent space, and then maps back to the hyperbolic space.

$x_1 y_1 + \ldots + x_d y_d$. We denote $\mathbb{H}^{d,K}$ as the hyperboloid manifold in $d$ dimensions with constant negative **curvature** $-1/K$ ($K > 0$), and $\mathcal{T}_{\mathbf{x}}\mathbb{H}^{d,K}$ the (Euclidean) **tangent space** centered at point $\mathbf{x}$

$$\mathbb{H}^{d,K} := \{\mathbf{x} \in \mathbb{R}^{d+1} : \langle \mathbf{x}, \mathbf{x} \rangle_{\mathcal{L}} = -K, x_0 > 0\} \quad \mathcal{T}_{\mathbf{x}}\mathbb{H}^{d,K} := \{\mathbf{v} \in \mathbb{R}^{d+1} : \langle \mathbf{v}, \mathbf{x} \rangle_{\mathcal{L}} = 0\}. \quad (3)$$

Now for $\mathbf{v}$ and $\mathbf{w}$ in $\mathcal{T}_{\mathbf{x}}\mathbb{H}^{d,K}$, $g_{\mathbf{x}}^K(\mathbf{v}, \mathbf{w}) := \langle \mathbf{v}, \mathbf{w} \rangle_{\mathcal{L}}$ is a Riemannian metric tensor [33] and $(\mathbb{H}^{d,K}, g_{\mathbf{x}}^K)$ is a Riemannian manifold with negative curvature $-1/K$. $\mathcal{T}_{\mathbf{x}}\mathbb{H}^{d,K}$ is a local, first-order approximation of the hyperbolic manifold at $\mathbf{x}$ and the restriction of the Minkowski inner product to $\mathcal{T}_{\mathbf{x}}\mathbb{H}^{d,K}$ is positive definite. $\mathcal{T}_{\mathbf{x}}\mathbb{H}^{d,K}$ is useful to perform Euclidean operations undefined in hyperbolic space and we denote $||\mathbf{v}||_{\mathcal{L}} = \sqrt{\langle \mathbf{v}, \mathbf{v} \rangle_{\mathcal{L}}}$ as the norm of $\mathbf{v} \in \mathcal{T}_{\mathbf{x}}\mathbb{H}^{d,K}$.

**Geodesics and induced distances**. Next, we introduce the notion of geodesics and distances in manifolds, which are generalizations of shortest paths in graphs or straight lines in Euclidean geometry (Figure 1). Geodesics and distance functions are particularly important in graph embedding algorithms, as a common optimization objective is to minimize geodesic distances between connected nodes. Let $\mathbf{x} \in \mathbb{H}^{d,K}$ and $\mathbf{u} \in \mathcal{T}_{\mathbf{x}}\mathbb{H}^{d,K}$, and assume that $\mathbf{u}$ is unit-speed, *i.e.* $\langle \mathbf{u}, \mathbf{u} \rangle_{\mathcal{L}} = 1$, then we have the following result:

**Proposition 3.1.** *Let* $\mathbf{x} \in \mathbb{H}^{d,K}$, $\mathbf{u} \in \mathcal{T}_{\mathbf{x}}\mathbb{H}^{d,K}$ *be unit-speed. The unique unit-speed geodesic* $\gamma_{\mathbf{x} \to \mathbf{u}}(\cdot)$ *such that* $\gamma_{\mathbf{x} \to \mathbf{u}}(0) = \mathbf{x}$, $\dot{\gamma}_{\mathbf{x} \to \mathbf{u}}(0) = \mathbf{u}$ *is* $\gamma_{\mathbf{x} \to \mathbf{u}}^K(t) = \cosh\left(\frac{t}{\sqrt{K}}\right)\mathbf{x} + \sqrt{K}\sinh\left(\frac{t}{\sqrt{K}}\right)\mathbf{u}$, *and the intrinsic distance function between two points* $\mathbf{x}, \mathbf{y}$ *in* $\mathbb{H}^{d,K}$ *is then:*

$$d_{\mathcal{L}}^K(\mathbf{x}, \mathbf{y}) = \sqrt{K}\operatorname{arcosh}(-\langle \mathbf{x}, \mathbf{y} \rangle_{\mathcal{L}}/K). \quad (4)$$

**Exponential and logarithmic maps**. Mapping between tangent space and hyperbolic space is done by exponential and logarithmic maps. Given $\mathbf{x} \in \mathbb{H}^{d,K}$ and a tangent vector $\mathbf{v} \in \mathcal{T}_{\mathbf{x}}\mathbb{H}^{d,K}$, the exponential map $\exp_{\mathbf{x}}^K : \mathcal{T}_{\mathbf{x}}\mathbb{H}^{d,K} \to \mathbb{H}^{d,K}$ assigns to $\mathbf{v}$ the point $\exp_{\mathbf{x}}^K(\mathbf{v}) := \gamma(1)$, where $\gamma$ is the unique geodesic satisfying $\gamma(0) = \mathbf{x}$ and $\dot{\gamma}(0) = \mathbf{v}$. The logarithmic map is the reverse map that maps back to the tangent space at $\mathbf{x}$ such that $\log_{\mathbf{x}}^K(\exp_{\mathbf{x}}^K(\mathbf{v})) = \mathbf{v}$. In general Riemannian manifolds, these operations are only defined locally but in the hyperbolic space, they form a bijection between the hyperbolic space and the tangent space at a point. We have the following direct expressions of the exponential and the logarithmic maps, which allow us to perform operations on points on the hyperboloid manifold by mapping them to tangent spaces and vice-versa:

**Proposition 3.2.** *For* $\mathbf{x} \in \mathbb{H}^{d,K}$, $\mathbf{v} \in \mathcal{T}_{\mathbf{x}}\mathbb{H}^{d,K}$ *and* $\mathbf{y} \in \mathbb{H}^{d,K}$ *such that* $\mathbf{v} \neq \mathbf{0}$ *and* $\mathbf{y} \neq \mathbf{x}$, *the exponential and logarithmic maps of the hyperboloid model are given by:*

$$\exp_{\mathbf{x}}^K(\mathbf{v}) = \cosh\left(\frac{||\mathbf{v}||_{\mathcal{L}}}{\sqrt{K}}\right)\mathbf{x} + \sqrt{K}\sinh\left(\frac{||\mathbf{v}||_{\mathcal{L}}}{\sqrt{K}}\right)\frac{\mathbf{v}}{||\mathbf{v}||_{\mathcal{L}}}, \quad \log_{\mathbf{x}}^K(\mathbf{y}) = d_{\mathcal{L}}^K(\mathbf{x}, \mathbf{y})\frac{\mathbf{y} + \frac{1}{K}\langle \mathbf{x}, \mathbf{y} \rangle_{\mathcal{L}}\mathbf{x}}{||\mathbf{y} + \frac{1}{K}\langle \mathbf{x}, \mathbf{y} \rangle_{\mathcal{L}}\mathbf{x}||_{\mathcal{L}}}.$$

## 4 Hyperbolic Graph Convolutional Networks

Here we introduce HGCN, a generalization of inductive GCNs in hyperbolic geometry that benefits from the expressiveness of both graph neural networks and hyperbolic embeddings. First, since input

features are often Euclidean, we derive a mapping from Euclidean features to hyperbolic space. Next, we derive two components of graph convolution: The analogs of Euclidean feature transformation and feature aggregation (Equations 1, 2) in the hyperboloid model. Finally, we introduce the HGCN algorithm with trainable curvature.

## 4.1 Mapping from Euclidean to hyperbolic spaces

HGCN first maps input features to the hyperboloid manifold via the $\exp$ map. Let $\mathbf{x}^{0,E} \in \mathbb{R}^d$ denote input Euclidean features. For instance, these features could be produced by pre-trained Euclidean neural networks. Let $\mathbf{o} := \{\sqrt{K}, 0, \ldots, 0\} \in \mathbb{H}^{d,K}$ denote the north pole (origin) in $\mathbb{H}^{d,K}$, which we use as a reference point to perform tangent space operations. We have $\langle (0, \mathbf{x}^{0,E}), \mathbf{o} \rangle = 0$. Therefore, we interpret $(0, \mathbf{x}^{0,E})$ as a point in $\mathcal{T}_{\mathbf{o}}\mathbb{H}^{d,K}$ and use Proposition 3.2 to map it to $\mathbb{H}^{d,K}$ with:

$$\mathbf{x}^{0,H} = \exp_{\mathbf{o}}^K((0, \mathbf{x}^{0,E})) = \left( \sqrt{K}\cosh\left(\frac{||\mathbf{x}^{0,E}||_2}{\sqrt{K}}\right), \sqrt{K}\sinh\left(\frac{||\mathbf{x}^{0,E}||_2}{\sqrt{K}}\right) \frac{\mathbf{x}^{0,E}}{||\mathbf{x}^{0,E}||_2} \right). \quad (5)$$

## 4.2 Feature transform in hyperbolic space

The feature transform in Equation 1 is used in GCN to map the embedding space of one layer to the next layer embedding space and capture large neighborhood structures. We now want to learn transformations of points on the hyperboloid manifold. However, there is no notion of vector space structure in hyperbolic space. We build upon Hyperbolic Neural Network (HNN) [10] and derive transformations in the hyperboloid model. The main idea is to leverage the $\exp$ and $\log$ maps in Proposition 3.2 so that we can use the tangent space $\mathcal{T}_{\mathbf{o}}\mathbb{H}^{d,K}$ to perform Euclidean transformations.

**Hyperboloid linear transform**. Linear transformation requires multiplication of the embedding vector by a weight matrix, followed by bias translation. To compute matrix vector multiplication, we first use the logarithmic map to project hyperbolic points $\mathbf{x}^H$ to $\mathcal{T}_{\mathbf{o}}\mathbb{H}^{d,K}$. Thus the matrix representing the transform is defined on the tangent space, which is Euclidean and isomorphic to $\mathbb{R}^d$. We then project the vector in the tangent space back to the manifold using the exponential map. Let $W$ be a $d' \times d$ weight matrix. We define the hyperboloid matrix multiplication as:

$$W \otimes^K \mathbf{x}^H := \exp_{\mathbf{o}}^K(W\log_{\mathbf{o}}^K(\mathbf{x}^H)), \quad (6)$$

where $\log_{\mathbf{o}}^K(\cdot)$ is on $\mathbb{H}^{d,K}$ and $\exp_{\mathbf{o}}^K(\cdot)$ maps to $\mathbb{H}^{d',K}$. In order to perform bias addition, we use a result from the HNN model and define $\mathbf{b}$ as an Euclidean vector located at $\mathcal{T}_{\mathbf{o}}\mathbb{H}^{d,K}$. We then parallel transport $\mathbf{b}$ to the tangent space of the hyperbolic point of interest and map it to the manifold. If $P_{\mathbf{o} \to \mathbf{x}^H}^K(\cdot)$ is the parallel transport from $\mathcal{T}_{\mathbf{o}}\mathbb{H}^{d',K}$ to $\mathcal{T}_{\mathbf{x}^H}\mathbb{H}^{d',K}$ (*c.f.* Appendix A for details), the hyperboloid bias addition is then defined as:

$$\mathbf{x}^H \oplus^K \mathbf{b} := \exp_{\mathbf{x}^H}^K(P_{\mathbf{o} \to \mathbf{x}^H}^K(\mathbf{b})). \quad (7)$$

## 4.3 Neighborhood aggregation on the hyperboloid manifold

Aggregation (Equation 2) is a crucial step in GCNs as it captures neighborhood structures and features. Suppose that $\mathbf{x}_i$ aggregates information from its neighbors $(\mathbf{x}_j)_{j \in \mathcal{N}(i)}$ with weights $(w_j)_{j \in \mathcal{N}(i)}$. Mean aggregation in Euclidean GCN computes the weighted average $\sum_{j \in \mathcal{N}(i)} w_j \mathbf{x}_j$. An analog of mean aggregation in hyperbolic space is the Fréchet mean [9], which, however, has no closed form solution. Instead, we propose to perform aggregation in tangent spaces using hyperbolic attention.

**Attention based aggregation**. Attention in GCNs learns a notion of neighbors' importance and aggregates neighbors' messages according to their importance to the center node. However, attention on Euclidean embeddings does not take into account the hierarchical nature of many real-world networks. Thus, we further propose hyperbolic attention-based aggregation. Given hyperbolic embeddings $(\mathbf{x}_i^H, \mathbf{x}_j^H)$, we first map $\mathbf{x}_i^H$ and $\mathbf{x}_j^H$ to the tangent space of the origin to compute attention weights $w_{ij}$ with concatenation and Euclidean Multi-layer Percerption (MLP). We then propose a hyperbolic aggregation to average nodes' representations:

$$w_{ij} = \text{SOFTMAX}_{j \in \mathcal{N}(i)}(\text{MLP}(\log_{\mathbf{o}}^K(\mathbf{x}_i^H)||\log_{\mathbf{o}}^K(\mathbf{x}_j^H))) \quad (8)$$

$$\text{AGG}^K(\mathbf{x}^H)_i = \exp_{\mathbf{x}_i^H}^K\left( \sum_{j \in \mathcal{N}(i)} w_{ij}\log_{\mathbf{x}_i^H}^K(\mathbf{x}_j^H) \right). \quad (9)$$

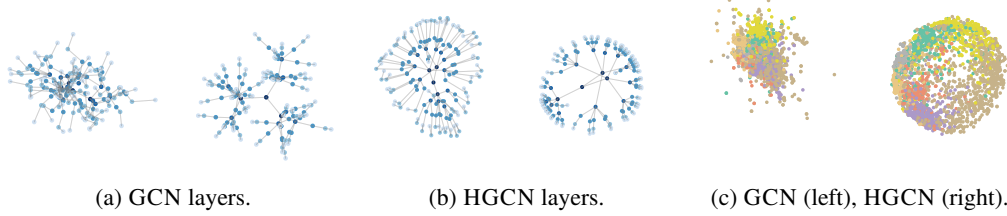

(a) GCN layers.　　　　　　(b) HGCN layers.　　　　　　(c) GCN (left), HGCN (right).

Figure 3: Visualization of embeddings for LP on DISEASE and NC on CORA (visualization on the Poincaré disk for HGCN). (a) GCN embeddings in first and last layers for DISEASE LP hardly capture hierarchy (depth indicated by color). (b) In contrast, HGCN preserves node hierarchies. (c) On CORA NC, HGCN leads to better class separation (indicated by different colors).

Note that our proposed aggregation is directly performed in the tangent space of each center point $\mathbf{x}_i^H$, as this is where the Euclidean approximation is best (*cf.* Figure 2). We show in our ablation experiments (*cf.* Table 2) that this local aggregation outperforms aggregation in tangent space at the origin ($\text{AGG}_{\mathbf{o}}$), due to the fact that relative distances have lower distortion in our approach.

**Non-linear activation with different curvatures**. Analogous to Euclidean aggregation (Equation 2), HGCN uses a non-linear activation function, $\sigma(\cdot)$ such that $\sigma(0) = 0$, to learn non-linear transformations. Given hyperbolic curvatures $-1/K_{\ell-1}, -1/K_\ell$ at layer $\ell - 1$ and $\ell$ respectively, we introduce a hyperbolic non-linear activation $\sigma^{\otimes^{K_{\ell-1},K_\ell}}$ with different curvatures. This step is crucial as it allows us to smoothly vary curvature at each layer. More concretely, HGCN applies the Euclidean non-linear activation in $\mathcal{T}_{\mathbf{o}}\mathbb{H}^{d,K_{\ell-1}}$ and then maps back to $\mathbb{H}^{d,K_\ell}$:

$$\sigma^{\otimes^{K_{\ell-1},K_\ell}}(\mathbf{x}^H) = \exp_{\mathbf{o}}^{K_\ell}(\sigma(\log_{\mathbf{o}}^{K_{\ell-1}}(\mathbf{x}^H))). \tag{10}$$

Note that in order to apply the exponential map, points must be located in the tangent space at the north pole. Fortunately, tangent spaces of the north pole are shared across hyperboloid manifolds of the same dimension that have different curvatures, making Equation 10 mathematically correct.

## 4.4　HGCN architecture

Having introduced all the building blocks of HGCN, we now summarize the model architecture. Given a graph $\mathcal{G} = (\mathcal{V}, \mathcal{E})$ and input Euclidean features $(\mathbf{x}^{0,E})_{i \in \mathcal{V}}$, the first layer of HGCN maps from Euclidean to hyperbolic space as detailed in Section 4.1. HGCN then stacks multiple hyperbolic graph convolution layers. At each layer HGCN transforms and aggregates neighbour's embeddings in the tangent space of the center node and projects the result to a hyperbolic space with different curvature. Hence the message passing in a HGCN layer is:

$$\mathbf{h}_i^{\ell,H} = (W^\ell \otimes^{K_{\ell-1}} \mathbf{x}_i^{\ell-1,H}) \oplus^{K_{\ell-1}} \mathbf{b}^\ell \qquad \text{(hyperbolic feature transform)} \tag{11}$$

$$\mathbf{y}_i^{\ell,H} = \text{AGG}^{K_{\ell-1}}(\mathbf{h}^{\ell,H})_i \qquad \text{(attention-based neighborhood aggregation)} \tag{12}$$

$$\mathbf{x}_i^{\ell,H} = \sigma^{\otimes^{K_{\ell-1},K_\ell}}(\mathbf{y}_i^{\ell,H}) \qquad \text{(non-linear activation with different curvatures)} \tag{13}$$

where $-1/K_{\ell-1}$ and $-1/K_\ell$ are the hyperbolic curvatures at layer $\ell-1$ and $\ell$ respectively. Hyperbolic embeddings $(\mathbf{x}^{L,H})_{i \in \mathcal{V}}$ at the last layer can then be used to predict node attributes or links.

For link prediction, we use the Fermi-Dirac decoder [23, 29], a generalization of sigmoid, to compute probability scores for edges:

$$p((i,j) \in \mathcal{E}|\mathbf{x}_i^{L,H}, \mathbf{x}_j^{L,H}) = \left[e^{(d_{\mathcal{L}}^{K_L}(\mathbf{x}_i^{L,H}, \mathbf{x}_j^{L,H})^2 - r)/t} + 1\right]^{-1}, \tag{14}$$

where $d_{\mathcal{L}}^{K_L}(\cdot, \cdot)$ is the hyperbolic distance and $r$ and $t$ are hyper-parameters. We then train HGCN by minimizing the cross-entropy loss using negative sampling.

For node classification, we map the output of the last HGCN layer to the tangent space of the origin with the logarithmic map $\log_{\mathbf{o}}^{K_L}(\cdot)$ and then perform Euclidean multinomial logistic regression. Note that another possibility is to directly classify points on the hyperboloid manifold using the hyperbolic multinomial logistic loss [10]. This method performs similarly to Euclidean classification (*cf.* [10] for an empirical comparison). Finally, we also add a link prediction regularization objective in node classification tasks, to encourage embeddings at the last layer to preserve the graph structure.

| Dataset | DISEASE | | DISEASE-M | | HUMAN PPI | | AIRPORT | | PUBMED | | CORA | |
| Hyperbolicity δ | δ = 0 | | δ = 0 | | δ = 1 | | δ = 1 | | δ = 3.5 | | δ = 11 | |
| Method | LP | NC | LP | NC | LP | NC | LP | NC | LP | NC | LP | NC |
|---|---|---|---|---|---|---|---|---|---|---|---|---|
| EUC | 59.8±2.0 | 32.5±1.1 | - | - | - | - | 92.0±0.0 | 60.9±3.4 | 83.3±0.1 | 48.2±0.7 | 82.5±0.3 | 23.8±0.7 |
| HYP [29] | 63.5±0.6 | 45.5±3.3 | - | - | - | - | 94.5±0.0 | 70.2±0.1 | 87.5±0.1 | 68.5±0.3 | 87.6±0.2 | 22.0±1.5 |
| EUC-MIXED | 49.6±1.1 | 35.2±3.4 | - | - | - | - | 91.5±0.1 | 68.3±2.3 | 86.0±1.3 | 63.0±0.3 | 84.4±0.2 | 46.1±0.4 |
| HYP-MIXED | 55.1±1.3 | 56.9±1.5 | - | - | - | - | 93.3±0.0 | 69.6±0.1 | 83.8±0.3 | 73.9±0.2 | 85.6±0.5 | 45.9±0.3 |
| MLP | 72.6±0.6 | 28.8±2.5 | 55.3±0.5 | 55.9±0.3 | 67.8±0.2 | 55.3±0.4 | 89.8±0.5 | 68.6±0.6 | 84.1±0.9 | 72.4±0.2 | 83.1±0.5 | 51.5±1.0 |
| HNN[10] | 75.1±0.3 | 41.0±1.8 | 60.9±0.4 | 56.2±0.3 | 72.9±0.3 | 59.3±0.4 | 90.8±0.2 | 80.5±0.5 | 94.9±0.1 | 69.8±0.4 | 89.0±0.1 | 54.6±0.4 |
| GCN[21] | 64.7±0.5 | 69.7±0.4 | 66.0±0.8 | 59.4±3.4 | 77.0±0.5 | 69.7±0.3 | 89.3±0.4 | 81.4±0.6 | 91.1±0.5 | 78.1±0.2 | 90.4±0.2 | 81.3±0.3 |
| GAT [41] | 69.8±0.3 | 70.4±0.4 | 69.5±0.4 | 62.5±0.7 | 76.8±0.4 | 70.5±0.4 | 90.5±0.3 | 81.5±0.3 | 91.2±0.1 | 79.0±0.3 | 93.7±0.1 | 83.0±0.7 |
| SAGE [15] | 65.9±0.3 | 69.1±0.6 | 67.4±0.5 | 61.3±0.4 | 78.1±0.6 | 69.1±0.3 | 90.4±0.5 | 82.1±0.5 | 86.2±1.0 | 77.4±2.2 | 85.5±0.6 | 77.9±2.4 |
| SGC [44] | 65.1±0.2 | 69.5±0.2 | 66.2±0.2 | 60.5±0.3 | 76.1±0.2 | 71.3±0.1 | 89.8±0.3 | 80.6±0.1 | 94.1±0.0 | 78.9±0.0 | 91.5±0.1 | 81.0±0.1 |
| HGCN | **90.8±0.3** | **74.5±0.9** | **78.1±0.4** | **72.2±0.5** | **84.5±0.4** | **74.6±0.3** | **96.4±0.1** | **90.6±0.2** | **96.3±0.0** | **80.3±0.3** | 92.9±0.1 | 79.9±0.2 |
| (%) ERR RED | -63.1% | -13.8% | -28.2% | -25.9% | -29.2% | -11.5% | -60.9% | -47.5% | -27.5% | -6.2% | +12.7% | +18.2% |

Table 1: ROC AUC for Link Prediction (LP) and F1 score for Node Classification (NC) tasks. For inductive datasets, we only evaluate inductive methods since shallow methods cannot generalize to unseen nodes/graphs. We report graph hyperbolicity values δ (lower is more hyperbolic).

## 4.5 Trainable curvature

We further analyze the effect of trainable curvatures in HGCN. Theorem 4.1 (proof in Appendix B) shows that assuming infinite precision, for the link prediction task, we can achieve the same performance for varying curvatures with an affine invariant decoder by scaling embeddings.

**Theorem 4.1.** *For any hyperbolic curvatures* $-1/K, -1/K' < 0$*, for any node embeddings* $H = \{\mathbf{h}_i\} \subset \mathbb{H}^{d,K}$ *of a graph* $G$*, we can find* $H' \subset \mathbb{H}^{d,K'}$*,* $H' = \{\mathbf{h}'_i | \mathbf{h}'_i = \sqrt{\frac{K'}{K}}\mathbf{h}_i\}$*, such that the reconstructed graph from* $H'$ *via the Fermi-Dirac decoder is the same as the reconstructed graph from* $H$*, with different decoder parameters* $(r, t)$ *and* $(r', t')$*.*

However, despite the same expressive power, adjusting curvature at every layer is important for good performance in practice due to factors of limited machine precision and normalization. First, with very low or very high curvatures, the scaling factor $\frac{K'}{K}$ in Theorem 4.1 becomes close to 0 or very large, and limited machine precision results in large error due to rounding. This is supported by Figure 4 and Table 2 where adjusting and training curvature lead to significant performance gain. Second, the norms of hidden layers that achieve the same local minimum in training also vary by a factor of $\sqrt{K}$. In practice, however, optimization is much more stable when the values are normalized [16]. In the context of HGCN, trainable curvature provides a natural way to learn embeddings of the right scale at each layer, improving optimization. Figure 4 shows the effect of decreasing curvature ($K = +\infty$ is the Euclidean case) on link prediction performance.

## 5 Experiments

We comprehensively evaluate our method on a variety of networks, on both node classification (NC) and link prediction (LP) tasks, in transductive and inductive settings. We compare performance of HGCN against a variety of shallow and GNN-based baselines. We further use visualizations to investigate the expressiveness of HGCN in link prediction tasks, and also demonstrate its ability to learn embeddings that capture the hierarchical structure of many real-world networks.

### 5.1 Experimental setup

**Datasets**. We use a variety of open transductive and inductive datasets that we detail below (more details in Appendix). We compute Gromovs $\delta-$hyperbolicity [1, 28, 17], a notion from group theory that measures how tree-like a graph is. The lower $\delta$, the more hyperbolic is the graph dataset, and $\delta = 0$ for trees. We conjecture that HGCN works better on graphs with small $\delta$-hyperbolicity.

1. **Citation networks**. CORA [36] and PUBMED [27] are standard benchmarks describing citation networks where nodes represent scientific papers, edges are citations between them, and node labels are academic (sub)areas. CORA contains 2,708 machine learning papers divided into 7 classes while PUBMED has 19,717 publications in the area of medicine grouped in 3 classes.
2. **Disease propagation tree**. We simulate the SIR disease spreading model [2], where the label of a node is whether the node was infected or not. Based on the model, we build tree networks, where

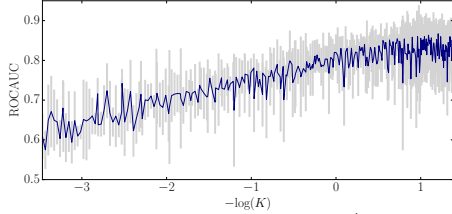

| Method | DISEASE | AIRPORT |
|---|---|---|
| HGCN | $78.4 \pm 0.3$ | $91.8 \pm 0.3$ |
| HGCN-ATT$_o$ | $80.9 \pm 0.4$ | $92.3 \pm 0.3$ |
| HGCN-ATT | $82.0 \pm 0.2$ | $92.5 \pm 0.2$ |
| HGCN-C | $89.1 \pm 0.2$ | $94.9 \pm 0.3$ |
| HGCN-ATT-C | $\mathbf{90.8} \pm 0.3$ | $\mathbf{96.4} \pm 0.1$ |

Figure 4: Decreasing curvature $(-1/K)$ improves link prediction performance on DISEASE.

Table 2: ROC AUC for link prediction on AIRPORT and DISEASE datasets.

node features indicate the susceptibility to the disease. We build transductive and inductive variants of this dataset, namely DISEASE and DISEASE-M (which contains multiple tree components).

3. **Protein-protein interactions (PPI) networks**. PPI is a dataset of human PPI networks [37]. Each human tissue has a PPI network, and the dataset is a union of PPI networks for human tissues. Each protein has a label indicating the stem cell growth rate after 19 days [40], which we use for the node classification task. The 16-dimensional feature for each node represents the RNA expression levels of the corresponding proteins, and we perform log transform on the features.

4. **Flight networks**. AIRPORT is a transductive dataset where nodes represent airports and edges represent the airline routes as from OpenFlights.org. Compared to previous compilations [49], our dataset has larger size (2,236 nodes). We also augment the graph with geographic information (longitude, latitude and altitude), and GDP of the country where the airport belongs to. We use the population of the country where the airport belongs to as the label for node classification.

**Baselines**. For shallow methods, we consider Euclidean embeddings (EUC) and Poincaré embeddings (HYP) [29]. We conjecture that HYP will outperform EUC on hierarchical graphs. For a fair comparison with HGCN which leverages node features, we also consider EUC-MIXED and HYP-MIXED baselines, where we concatenate the corresponding shallow embeddings with node features, followed by a MLP to predict node labels or links. For state-of-the-art Euclidean GNN models, we consider GCN [21], GraphSAGE (SAGE) [15], Graph Attention Networks (GAT) [41] and Simplified Graph Convolution (SGC) [44][3]. We also consider feature-based approaches: MLP and its hyperbolic variant (HNN) [10], which does not utilize the graph structure.

**Training**. For all methods, we perform a hyper-parameter search on a validation set over initial learning rate, weight decay, dropout[4], number of layers, and activation functions. We measure performance on the final test set over 10 random parameter initializations. For fairness, we also control the number of dimensions to be the same (16) for all methods. We optimize all models with Adam [19], except Poincaré embeddings which are optimized with RiemannianSGD [4, 48]. Further details can be found in Appendix. We open source our implementation[5] of HGCN and baselines.

**Evaluation metric**. In transductive LP tasks, we randomly split edges into $85/5/10\%$ for training, validation and test sets. For transductive NC, we use $70/15/15\%$ splits for AIRPORT, $30/10/60\%$ splits for DISEASE, and we use standard splits [21, 46] with 20 train examples per class for CORA and PUBMED. One of the main advantages of HGCN over related hyperbolic graph embedding is its inductive capability. For inductive tasks, the split is performed across graphs. All nodes/edges in training graphs are considered the training set, and the model is asked to predict node class or unseen links for test graphs. Following previous works, we evaluate link prediction by measuring area under the ROC curve on the test set and evaluate node classification by measuring F1 score, except for CORA and PUBMED, where we report accuracy as is standard in the literature.

## 5.2 Results

Table 1 reports the performance of HGCN in comparison to baseline methods. HGCN works best in inductive scenarios where both node features and network topology play an important role.

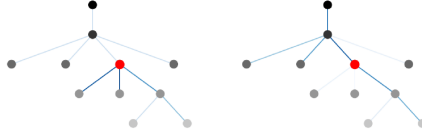

Figure 5: Attention: Euclidean GAT (left), HGCN (right). Each graph represents a 2-hop neighborhood of the DISEASE-M dataset.

The performance gain of HGCN with respect to Euclidean GNN models is correlated with graph hyperbolicity. HGCN achieves an average of 45.4% (LP) and 12.3% (NC) error reduction compared to the best deep baselines for graphs with high hyperbolicity (low $\delta$), suggesting that GNNs can significantly benefit from hyperbolic geometry, especially in link prediction tasks. Furthermore, the performance gap between HGCN and HNN suggests that neighborhood aggregation has been effective in learning node representations in graphs. For example, in disease spread datasets, both Euclidean attention and hyperbolic geometry lead to significant improvement of HGCN over other baselines. This can be explained by the fact that in disease spread trees, parent nodes contaminate their children. HGCN can successfully model these asymmetric and hierarchical relationships with hyperbolic attention and improves performance over all baselines.

On the CORA dataset with low hyperbolicity, HGCN does not outperform Euclidean GNNs, suggesting that Euclidean geometry is better for its underlying graph structure. However, for small dimensions, HGCN is still significantly more effective than GCN even with CORA. Figure 3c shows 2-dimensional HGCN and GCN embeddings trained with LP objective, where colors denote the label class. HGCN achieves much better label class separation.

### 5.3 Analysis

**Ablations**. We further analyze the effect of proposed components in HGCN, namely hyperbolic attention (ATT) and trainable curvature (C) on AIRPORT and DISEASE datasets in Table 2. We observe that both attention and trainable curvature lead to performance gains over HGCN with fixed curvature and no attention. Furthermore, our attention model ATT outperforms $\text{ATT}_{\mathbf{o}}$ (aggregation in tangent space at $\mathbf{o}$), and we conjecture that this is because the local Euclidean average is a better approximation near the center point rather than near $\mathbf{o}$. Finally, the addition of both ATT and C improves performance even further, suggesting that both components are important in HGCN.

**Visualizations**. We first visualize the GCN and HGCN embeddings at the first and last layers in Figure 3. We train HGCN with 3-dimensional hyperbolic embeddings and map them to the Poincaré disk which is better for visualization. In contrast to GCN, tree structure is preserved in HGCN, where nodes close to the center are higher in the hierarchy of the tree. This way HGCN smoothly transforms Euclidean features to Hyperbolic embeddings that preserve node hierarchy.

Figure 5 shows the attention weights in the 2-hop neighborhood of a center node (red) for the DISEASE dataset. The red node is the node where we compute attention. The darkness of the color for other nodes denotes their hierarchy. The attention weights for nodes in the neighborhood are visualized by the intensity of edges. We observe that in HGCN the center node pays more attention to its (grand)parent. In contrast to Euclidean GAT, our aggregation with attention in hyperbolic space allows us to pay more attention to nodes with high hierarchy. Such attention is crucial to good performance in DISEASE, because only sick parents will propagate the disease to their children.

## 6 Conclusion

We introduced HGCN, a novel architecture that learns hyperbolic embeddings using graph convolutional networks. In HGCN, the Euclidean input features are successively mapped to embeddings in hyperbolic spaces with trainable curvatures at every layer. HGCN achieves new state-of-the-art in learning embeddings for real-world hierarchical and scale-free graphs.

## Acknowledgments

Jure Leskovec is a Chan Zuckerberg Biohub investigator. This research has been supported in part by DARPA under FA865018C7880 (ASED), (MSC); NIH under No. U54EB020405 (Mobilize); ARO under MURI; IARPA under No. 2017-17071900005 (HFC), NSF under No. OAC-1835598 (CINES); Stanford Data Science Initiative, Chan Zuckerberg Biohub, JD.com, Amazon, Boeing, Docomo, Huawei, Hitachi, Observe, Siemens, and UST Global. We gratefully acknowledge the support of DARPA under Nos. FA87501720095 (D3M), FA86501827865 (SDH), and FA86501827882 (ASED); NIH under No. U54EB020405 (Mobilize), NSF under Nos. CCF1763315 (Beyond Sparsity), CCF1563078 (Volume to Velocity), and 1937301 (RTML); ONR under No. N000141712266 (Unifying Weak Supervision); the Moore Foundation, NXP, Xilinx, LETI-CEA, Intel, IBM, Microsoft, NEC, Toshiba, TSMC, ARM, Hitachi, BASF, Accenture, Ericsson, Qualcomm, Analog Devices, the Okawa Foundation, American Family Insurance, Google Cloud, Swiss Re, TOTAL, and members of the Stanford DAWN project: Teradata, Facebook, Google, Ant Financial, NEC, VMWare, and Infosys. The U.S. Government is authorized to reproduce and distribute reprints for Governmental purposes notwithstanding any copyright notation thereon. Any opinions, findings, and conclusions or recommendations expressed in this material are those of the authors and do not necessarily reflect the views, policies, or endorsements, either expressed or implied, of DARPA, NIH, ONR, or the U.S. Government.

## Footnotes

[2]Project website with code and data: http://snap.stanford.edu/hgcn

[39]. However, shallow (Euclidean and hyperbolic) embedding methods have three major downsides: (1) They fail to leverage rich node feature information, which can be crucial in tasks such as node classification. (2) These methods are transductive, and therefore cannot be used for inference on unseen graphs. And, (3) they scale poorly as the number of model parameters grows linearly with the number of nodes.

[3]The equivalent of GCN in link prediction is GAE [20]. We did not compare link prediction GNNs based on shallow embeddings such as [49] since they are not inductive.

[4]HGCN uses DropConnect [42], as described in Appendix C.

[5]Code available at http://snap.stanford.edu/hgcn. We provide HGCN implementations for hyperboloid and Poincaré models. Empirically, both models give similar performance but hyperboloid model offers more stable optimization, because Poincaré distance is numerically unstable [30].

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
