[Supplementary Material]

# A  Review of Differential Geometry

We first recall some definitions of differential and hyperbolic geometry.

## A.1  Differential geometry

**Manifold**. An $d-$dimensional *manifold* $\mathcal{M}$ is a topological space that locally resembles the topological space $\mathbb{R}^d$ near each point. More concretely, for each point $\mathbf{x}$ on $\mathcal{M}$, we can find a *homeomorphism* (continuous bijection with continuous inverse) between a neighbourhood of $\mathbf{x}$ and $\mathbb{R}^d$. The notion of manifold is a generalization of surfaces in high dimensions.

**Tangent space**. Intuitively, if we think of $\mathcal{M}$ as a $d-$dimensional manifold embedded in $\mathbb{R}^{d+1}$, the *tangent space* $\mathcal{T}_{\mathbf{x}}\mathcal{M}$ at point $\mathbf{x}$ on $\mathcal{M}$ is a $d-$dimensional hyperplane in $\mathbb{R}^{d+1}$ that best approximates $\mathcal{M}$ around $\mathbf{x}$. Another possible interpretation for $\mathcal{T}_{\mathbf{x}}\mathcal{M}$ is that it contains all the possible directions of curves on $\mathcal{M}$ passing through $\mathbf{x}$. The elements of $\mathcal{T}_{\mathbf{x}}\mathcal{M}$ are called *tangent vectors* and the union of all tangent spaces is called the *tangent bundle* $\mathcal{T}\mathcal{M} = \cup_{\mathbf{x}\in\mathcal{M}}\mathcal{T}_{\mathbf{x}}\mathcal{M}$.

**Riemannian manifold**. A *Riemannian manifold* is a pair $(\mathcal{M}, \mathbf{g})$, where $\mathcal{M}$ is a smooth manifold and $\mathbf{g} = (g_{\mathbf{x}})_{\mathbf{x}\in\mathcal{M}}$ is a *Riemannian metric*, that is a family of smoothly varying inner products on tangent spaces, $g_{\mathbf{x}} : \mathcal{T}_{\mathbf{x}}\mathcal{M} \times \mathcal{T}_{\mathbf{x}}\mathcal{M} \to \mathbb{R}$. Riemannian metrics can be used to measure distances on manifolds.

**Distances and geodesics**. Let $(\mathcal{M}, \mathbf{g})$ be a Riemannian manifold. For $\mathbf{v} \in \mathcal{T}_{\mathbf{x}}\mathcal{M}$, define the norm of $\mathbf{v}$ by $||\mathbf{v}||_{\mathbf{g}} := \sqrt{g_{\mathbf{x}}(\mathbf{v}, \mathbf{v})}$. Suppose $\gamma : [a, b] \to \mathcal{M}$ is a smooth curve on $\mathcal{M}$. Define the length of $\gamma$ by:

$$L(\gamma) := \int_a^b ||\gamma'(t)||_{\mathbf{g}} dt.$$

Now with this definition of length, every connected Riemannian manifold becomes a metric space and the *distance* $d : \mathcal{M} \times \mathcal{M} \to [0, \infty)$ is defined as:

$$d(\mathbf{x}, \mathbf{y}) := \inf_{\gamma}\{L(\gamma) : \gamma \text{ is a continuously differentiable curve joining } \mathbf{x} \text{ and } \mathbf{y}\}.$$

*Geodesic* distances are a generalization of straight lines (or shortest paths) to non-Euclidean geometry. A curve $\gamma : [a, b] \to \mathcal{M}$ is *geodesic* if $d(\gamma(t), \gamma(s)) = L(\gamma|_{[t,s]})\forall(t, s) \in [a, b](t < s)$.

**Parallel transport**. *Parallel transport* is a generalization of translation to non-Euclidean geometry. Given a smooth manifold $\mathcal{M}$, parallel transport $P_{\mathbf{x}\to\mathbf{y}}(\cdot)$ maps a vector $\mathbf{v} \in \mathcal{T}_{\mathbf{x}}\mathcal{M}$ to $P_{\mathbf{x}\to\mathbf{y}}(\mathbf{v}) \in \mathcal{T}_{\mathbf{y}}\mathcal{M}$. In Riemannian geometry, parallel transport preserves the Riemannian metric tensor (norm, inner products...).

**Curvature**. At a high level, curvature measures how much a geometric object such as surfaces deviate from a flat plane. For instance, the Euclidean space has zero curvature while spheres have positive curvature. We illustrate the concept of curvature in Figure 6.

## A.2  Hyperbolic geometry

**Hyperbolic space**. The hyperbolic space in $d$ dimensions is the unique complete, simply connected $d-$dimensional Riemannian manifold with constant negative sectional curvature. There exist several models of hyperbolic space such as the Poincaré model or the hyperboloid model (also known as the Minkowski model or the Lorentz model). In what follows, we review the Poincaré and the hyperboloid models of hyperbolic space as well as connections between these two models.

### A.2.1  Poincaré ball model

Let $||.||_2$ be the Euclidean norm. The Poincaré ball model with unit radius and constant negative curvature $-1$ in $d$ dimensions is the Riemannian manifold $(\mathbb{D}^{d,1}, (g_{\mathbf{x}})_{\mathbf{x}})$ where

$$\mathbb{D}^{d,1} := \{\mathbf{x} \in \mathbb{R}^d : ||\mathbf{x}||^2 < 1\},$$

and

$$g_{\mathbf{x}} = \lambda_{\mathbf{x}}^2 I_d,$$

Figure 6: From left to right: a surface of negative curvature, a surface of zero curvature, and a surface of positive curvature.

where $\lambda_{\mathbf{x}} := \frac{2}{1-||\mathbf{x}||_2^2}$ and $I_d$ is the identity matrix. The induced distance between two points $(\mathbf{x}, \mathbf{y})$ in $\mathbb{D}^{d,1}$ can be computed as:

$$d_{\mathbb{D}}^1(\mathbf{x}, \mathbf{y}) = \text{arcosh}\left(1 + 2\frac{||\mathbf{x} - \mathbf{y}||_2^2}{(1 - ||\mathbf{x}||_2^2)(1 - ||\mathbf{y}||_2^2)}\right).$$

### A.2.2 Hyperboloid model

**Hyperboloid model**. Let $\langle ., .\rangle_{\mathcal{L}} : \mathbb{R}^{d+1} \times \mathbb{R}^{d+1} \to \mathbb{R}$ denote the Minkowski inner product,

$$\langle \mathbf{x}, \mathbf{y}\rangle_{\mathcal{L}} := -x_0 y_0 + x_1 y_1 + \ldots + x_d y_d.$$

The hyperboloid model with unit imaginary radius and constant negative curvature $-1$ in $d$ dimensions is defined as the Riemannian manifold $(\mathbb{H}^{d,1}, (g_{\mathbf{x}})_{\mathbf{x}})$ where

$$\mathbb{H}^{d,1} := \{\mathbf{x} \in \mathbb{R}^{d+1} : \langle \mathbf{x}, \mathbf{x}\rangle_{\mathcal{L}} = -1, x_0 > 0\},$$

and

$$g_{\mathbf{x}} := \begin{bmatrix} -1 & & & \\ & 1 & & \\ & & \ddots & \\ & & & 1 \end{bmatrix}.$$

The induced distance between two points $(\mathbf{x}, \mathbf{y})$ in $\mathbb{H}^{d,1}$ can be computed as:

$$d_{\mathcal{L}}^1(\mathbf{x}, \mathbf{y}) = \text{arcosh}(-\langle \mathbf{x}, \mathbf{y}\rangle_{\mathcal{L}}).$$

**Geodesics**. We recall a result that gives the unit speed geodesics in the hyperboloid model with curvature $-1$ [33]. This result can be used to show Propositions 3.1 and 3.2 for the hyperboloid manifold with negative curvature $-1/K$, and then learn $K$ as a model parameter in HGCN.

**Theorem A.1.** *Let $\mathbf{x} \in \mathbb{H}^{d,1}$ and $\mathbf{u} \in \mathcal{T}_{\mathbf{x}}\mathbb{H}^{d,1}$ unit-speed (i.e. $\langle \mathbf{u}, \mathbf{u}\rangle_{\mathcal{L}} = 1$). The unique unit-speed geodesic $\gamma_{\mathbf{x} \to \mathbf{u}} : [0, 1] \to \mathbb{H}^{d,1}$ such that $\gamma_{\mathbf{x} \to \mathbf{u}}(0) = \mathbf{x}$ and $\dot{\gamma}_{\mathbf{x} \to \mathbf{u}}(0) = \mathbf{u}$ is given by:*

$$\gamma_{\mathbf{x} \to \mathbf{u}}(t) = \cosh(t)\mathbf{x} + \sinh(t)\mathbf{u}.$$

**Parallel Transport**. If two points $\mathbf{x}$ and $\mathbf{y}$ on the hyperboloid $\mathbb{H}^{d,1}$ are connected by a geodesic, then the parallel transport of a tangent vector $\mathbf{v} \in \mathcal{T}_{\mathbf{x}}\mathbb{H}^{d,1}$ to the tangent space $\mathcal{T}_{\mathbf{y}}\mathbb{H}^{d,1}$ is:

$$P_{\mathbf{x} \to \mathbf{y}}(\mathbf{v}) = \mathbf{v} - \frac{\langle \log_{\mathbf{x}}(\mathbf{y}), \mathbf{v}\rangle_{\mathcal{L}}}{d_{\mathcal{L}}^1(\mathbf{x}, \mathbf{y})^2}(\log_{\mathbf{x}}(\mathbf{y}) + \log_{\mathbf{y}}(\mathbf{x})). \tag{15}$$

**Projections**. Finally, we recall projections to the hyperboloid manifold and its corresponding tangent spaces. A point $\mathbf{x} = (x_0, \mathbf{x}_{1:d}) \in \mathbb{R}^{d+1}$ can be projected on the hyperboloid manifold $\mathbb{H}^{d,1}$ with:

$$\Pi_{\mathbb{R}^{d+1} \to \mathbb{H}^{d,1}}(\mathbf{x}) := (\sqrt{1 + ||\mathbf{x}_{1:d}||_2^2}, \mathbf{x}_{1:d}). \tag{16}$$

Figure 7: Illustration of the hyperboloid model (top) in 3 dimensions and its connection to the Poincaré disk (bottom).

Similarly, a point $\mathbf{v} \in \mathbb{R}^{d+1}$ can be projected on $\mathcal{T}_{\mathbf{x}}\mathbb{H}^{d,1}$ with:

$$\Pi_{\mathbb{R}^{d+1} \to \mathcal{T}_{\mathbf{x}}\mathbb{H}^{d,1}}(\mathbf{v}) := \mathbf{v} + \langle \mathbf{x}, \mathbf{v} \rangle_{\mathcal{L}} \mathbf{x}. \tag{17}$$

In practice, these projections are very useful for optimization purposes as they constrain embeddings and tangent vectors to remain on the manifold and tangent spaces.

### A.2.3 Connection between the Poincaré ball model and the hyperboloid model

While the hyperboloid model tends to be more stable for optimization than the Poincaré model [30], the Poincaré model is very interpretable and embeddings can be directly visualized on the Poincaré disk. Fortunately, these two models are isomorphic (*cf.* Figure 7) and there exist a diffeomorphism $\Pi_{\mathbb{H}^{d,1} \to \mathbb{D}^{d,1}}(\cdot)$ mapping one space onto the other:

$$\Pi_{\mathbb{H}^{d,1} \to \mathbb{D}^{d,1}}(x_0, \ldots, x_d) = \frac{(x_1, \ldots, x_d)}{x_0 + 1} \tag{18}$$

$$\text{and} \quad \Pi_{\mathbb{D}^{d,1} \to \mathbb{H}^{d,1}}(x_1, \ldots, x_d) = \frac{(1 + ||\mathbf{x}||_2^2, 2x_1, \ldots, 2x_d)}{1 - ||\mathbf{x}||_2^2}. \tag{19}$$

## B  Proofs of Results

### B.1  Hyperboloid model of hyperbolic space

For completeness, we re-derive results of hyperbolic geometry for any arbitrary curvature. Similar derivations can be found in the literature [43].

**Proposition 3.1.** *Let* $\mathbf{x} \in \mathbb{H}^{d,K}$, $\mathbf{u} \in \mathcal{T}_{\mathbf{x}}\mathbb{H}^{d,K}$ *be unit-speed. The unique unit-speed geodesic* $\gamma_{\mathbf{x} \to \mathbf{u}}(\cdot)$ *such that* $\gamma_{\mathbf{x} \to \mathbf{u}}(0) = \mathbf{x}$, $\dot{\gamma}_{\mathbf{x} \to \mathbf{u}}(0) = \mathbf{u}$ *is* $\gamma^K_{\mathbf{x} \to \mathbf{u}}(t) = \cosh\left(\frac{t}{\sqrt{K}}\right)\mathbf{x} + \sqrt{K}\sinh\left(\frac{t}{\sqrt{K}}\right)\mathbf{u}$, *and the intrinsic distance function between two points* $\mathbf{x}, \mathbf{y}$ *in* $\mathbb{H}^{d,K}$ *is then:*

$$d^K_{\mathcal{L}}(\mathbf{x}, \mathbf{y}) = \sqrt{K}\operatorname{arcosh}(-\langle \mathbf{x}, \mathbf{y} \rangle_{\mathcal{L}}/K). \tag{4}$$

*Proof.* Using theorem A.1, we know that the unique unit-speed geodesic $\gamma_{\mathbf{y} \to \mathbf{u}}(.)$ in $\mathbb{H}^{d,1}$ must satisfy

$$\gamma_{\mathbf{y} \to \mathbf{u}}(0) = \mathbf{y} \text{ and } \dot{\gamma}_{\mathbf{y} \to \mathbf{u}}(0) = \mathbf{u} \text{ and } \frac{d}{dt}\langle \dot{\gamma}_{\mathbf{y} \to \mathbf{u}}(t), \dot{\gamma}_{\mathbf{y} \to \mathbf{u}}(t) \rangle_{\mathcal{L}} = 0 \ \forall t,$$

and is given by

$$\gamma_{\mathbf{y} \to \mathbf{u}}(t) = \cosh(t)\mathbf{y} + \sinh(t)\mathbf{u}.$$

Now let $\mathbf{x} \in \mathbb{H}^{d,K}$ and $\mathbf{u} \in \mathcal{T}_{\mathbf{x}}\mathbb{H}^{d,K}$ be unit-speed and denote $\gamma_{\mathbf{x}\to\mathbf{u}}^{K}(.)$ the unique unit-speed geodesic in $\mathbb{H}^{d,K}$ such that $\gamma_{\mathbf{x}\to\mathbf{u}}^{K}(0) = \mathbf{x}$ and $\dot{\gamma}_{\mathbf{x}\to\mathbf{u}}^{K}(0) = \mathbf{u}$. Let us define $\mathbf{y} := \frac{\mathbf{x}}{\sqrt{K}} \in \mathbb{H}^{d,1}$ and $\phi_{\mathbf{y}\to\mathbf{u}}(t) = \frac{1}{\sqrt{K}}\gamma_{\mathbf{x}\to\mathbf{u}}^{K}(\sqrt{K}t)$. We have,

$$\phi_{\mathbf{y}\to\mathbf{u}}(0) = \mathbf{y} \text{ and } \dot{\phi}_{\mathbf{y}\to\mathbf{u}}(0) = \mathbf{u},$$

and since $\gamma_{\mathbf{x}\to\mathbf{u}}^{K}(.)$ is the unique unit-speed geodesic in $\mathbb{H}^{d,K}$, we also have

$$\frac{d}{dt}\langle \dot{\phi}_{\mathbf{y}\to\mathbf{u}}(t), \dot{\phi}_{\mathbf{y}\to\mathbf{u}}(t)\rangle_{\mathcal{L}} = 0 \ \forall t.$$

Furthermore, we have $\mathbf{y} \in \mathbb{H}^{d,1}$, $\mathbf{u} \in \mathcal{T}_{\mathbf{y}}\mathbb{H}^{d,1}$ as $\langle \mathbf{u}, \mathbf{y}\rangle_{\mathcal{L}} = \frac{1}{\sqrt{K}}\langle \mathbf{u}, \mathbf{x}\rangle_{\mathcal{L}} = 0$ and $\langle \phi_{\mathbf{y}\to\mathbf{u}}(t), \phi_{\mathbf{y}\to\mathbf{u}}(t)\rangle_{\mathcal{L}} = -1 \forall t$. Therefore $\phi_{\mathbf{y}\to\mathbf{u}}(.)$ is a unit-speed geodesic in $\mathbb{H}^{d,1}$ and we get

$$\phi_{\mathbf{y}\to\mathbf{u}}(t) = \cosh(t)\mathbf{y} + \sinh(t)\mathbf{u}.$$

Finally, this leads to

$$\gamma_{\mathbf{x}\to\mathbf{u}}^{K}(t) = \cosh(\frac{t}{\sqrt{K}})\mathbf{x} + \sqrt{K}\sinh(\frac{t}{\sqrt{K}})\mathbf{u}.$$

$\square$

**Proposition 3.2.** *For $\mathbf{x} \in \mathbb{H}^{d,K}$, $\mathbf{v} \in \mathcal{T}_{\mathbf{x}}\mathbb{H}^{d,K}$ and $\mathbf{y} \in \mathbb{H}^{d,K}$ such that $\mathbf{v} \neq \mathbf{0}$ and $\mathbf{y} \neq \mathbf{x}$, the exponential and logarithmic maps of the hyperboloid model are given by:*

$$\exp_{\mathbf{x}}^{K}(\mathbf{v}) = \cosh\left(\frac{||\mathbf{v}||_{\mathcal{L}}}{\sqrt{K}}\right)\mathbf{x} + \sqrt{K}\sinh\left(\frac{||\mathbf{v}||_{\mathcal{L}}}{\sqrt{K}}\right)\frac{\mathbf{v}}{||\mathbf{v}||_{\mathcal{L}}}, \quad \log_{\mathbf{x}}^{K}(\mathbf{y}) = d_{\mathcal{L}}^{K}(\mathbf{x}, \mathbf{y})\frac{\mathbf{y} + \frac{1}{K}\langle \mathbf{x}, \mathbf{y}\rangle_{\mathcal{L}}\mathbf{x}}{||\mathbf{y} + \frac{1}{K}\langle \mathbf{x}, \mathbf{y}\rangle_{\mathcal{L}}\mathbf{x}||_{\mathcal{L}}}.$$

*Proof.* We use a similar reasoning to that in Corollary 1.1 in [11]. Let $\gamma_{\mathbf{x}\to\mathbf{v}}^{K}(.)$ be the unique geodesic such that $\gamma_{\mathbf{x}\to\mathbf{v}}^{K}(0) = \mathbf{x}$ and $\dot{\gamma}_{\mathbf{x}\to\mathbf{v}}^{K}(0) = \mathbf{v}$. Let us define $\mathbf{u} := \frac{\mathbf{v}}{||\mathbf{v}||_{\mathcal{L}}}$ where $||\mathbf{v}||_{\mathcal{L}} = \sqrt{\langle \mathbf{v}, \mathbf{v}\rangle_{\mathcal{L}}}$ is the Minkowski norm of $\mathbf{v}$ and

$$\phi_{\mathbf{x}\to\mathbf{u}}^{K}(t) := \gamma_{\mathbf{x}\to\mathbf{v}}^{K}\left(\frac{t}{||\mathbf{v}||_{\mathcal{L}}}\right).$$

$\phi_{\mathbf{x}\to\mathbf{u}}(t)$ satisfies,

$$\phi_{\mathbf{x}\to\mathbf{u}}^{K}(0) = \mathbf{x} \text{ and } \dot{\phi}_{\mathbf{x}\to\mathbf{u}}^{K}(0) = \mathbf{u} \text{ and } \frac{d}{dt}\langle \dot{\phi}_{\mathbf{x}\to\mathbf{u}}^{K}(t), \dot{\phi}_{\mathbf{x}\to\mathbf{u}}^{K}(t)\rangle_{\mathcal{L}} = 0 \ \forall t.$$

Therefore $\phi_{\mathbf{x}\to\mathbf{u}}^{K}(.)$ is a unit-speed geodesic in $\mathbb{H}^{d,K}$ and we get

$$\phi_{\mathbf{x}\to\mathbf{u}}^{K}(t) = \cosh(\frac{t}{\sqrt{K}})\mathbf{x} + \sqrt{K}\sinh(\frac{t}{\sqrt{K}})\mathbf{u}.$$

By identification, this leads to

$$\gamma_{\mathbf{x}\to\mathbf{v}}^{K}(t) = \cosh\left(\frac{||\mathbf{v}||_{\mathcal{L}}}{\sqrt{K}}t\right)\mathbf{x} + \sqrt{K}\sinh\left(\frac{||\mathbf{v}||_{\mathcal{L}}}{\sqrt{K}}t\right)\frac{\mathbf{v}}{||\mathbf{v}||_{\mathcal{L}}}.$$

We can use this result to derive exponential and logarthimic maps on the hyperboloid model. We know that $\exp_{\mathbf{x}}^{K}(\mathbf{v}) = \gamma_{\mathbf{x}\to\mathbf{v}}^{K}(1)$. Therefore we get,

$$\exp_{\mathbf{x}}^{K}(\mathbf{v}) = \cosh\left(\frac{||\mathbf{v}||_{\mathcal{L}}}{\sqrt{K}}\right)\mathbf{x} + \sqrt{K}\sinh\left(\frac{||\mathbf{v}||_{\mathcal{L}}}{\sqrt{K}}\right)\frac{\mathbf{v}}{||\mathbf{v}||_{\mathcal{L}}}.$$

Now let $\mathbf{y} = \exp_{\mathbf{x}}^{K}(\mathbf{v})$. We have $\langle \mathbf{x}, \mathbf{y}\rangle_{\mathcal{L}} = -K\cosh\left(\frac{||\mathbf{v}||_{\mathcal{L}}}{\sqrt{K}}\right)$ as $\langle \mathbf{x}, \mathbf{x}\rangle_{\mathcal{L}} = -K$ and $\langle \mathbf{x}, \mathbf{v}\rangle_{\mathcal{L}} = 0$. Therefore $\mathbf{y} + \frac{1}{K}\langle \mathbf{x}, \mathbf{y}\rangle_{\mathcal{L}}\mathbf{x} = \sqrt{K}\sinh\left(\frac{||\mathbf{v}||_{\mathcal{L}}}{\sqrt{K}}\right)\frac{\mathbf{v}}{||\mathbf{v}||_{\mathcal{L}}}$ and we get

$$\mathbf{v} = \sqrt{K}\operatorname{arsinh}\left(\frac{||\mathbf{y} + \frac{1}{K}\langle \mathbf{x}, \mathbf{y}\rangle_{\mathcal{L}}\mathbf{x}||_{\mathcal{L}}}{\sqrt{K}}\right)\frac{\mathbf{y} + \frac{1}{K}\langle \mathbf{x}, \mathbf{y}\rangle_{\mathcal{L}}\mathbf{x}}{||\mathbf{y} + \frac{1}{K}\langle \mathbf{x}, \mathbf{y}\rangle_{\mathcal{L}}\mathbf{x}||_{\mathcal{L}}},$$

where $||\mathbf{y} + \frac{1}{K}\langle\mathbf{x},\mathbf{y}\rangle_{\mathcal{L}}||_{\mathcal{L}}$ is well defined since $\mathbf{y} + \frac{1}{K}\langle\mathbf{x},\mathbf{y}\rangle_{\mathcal{L}}\mathbf{x} \in \mathcal{T}_{\mathbf{x}}\mathbb{H}^{d,K}$. Note that,

$$
\begin{aligned}
||\mathbf{y} + \frac{1}{K}\langle\mathbf{x},\mathbf{y}\rangle_{\mathcal{L}}\mathbf{x}||_{\mathcal{L}} &= \sqrt{\langle\mathbf{y},\mathbf{y}\rangle_{\mathcal{L}} + \frac{2}{K}\langle\mathbf{x},\mathbf{y}\rangle_{\mathcal{L}}^2 + \frac{1}{K^2}\langle\mathbf{x},\mathbf{y}\rangle_{\mathcal{L}}^2\langle\mathbf{x},\mathbf{x}\rangle_{\mathcal{L}}} \\
&= \sqrt{-K + \frac{1}{K}\langle\mathbf{x},\mathbf{y}\rangle_{\mathcal{L}}^2} \\
&= \sqrt{K}\sqrt{\langle\frac{\mathbf{x}}{\sqrt{K}},\frac{\mathbf{y}}{\sqrt{K}}\rangle_{\mathcal{L}}^2 - 1} \\
&= \sqrt{K}\sinh\operatorname{arcosh}\left(-\langle\frac{\mathbf{x}}{\sqrt{K}},\frac{\mathbf{y}}{\sqrt{K}}\rangle_{\mathcal{L}}\right)
\end{aligned}
$$

as $\langle\frac{\mathbf{x}}{\sqrt{K}},\frac{\mathbf{y}}{\sqrt{K}}\rangle_{\mathcal{L}} \leq -1$. Therefore, we finally have

$$
\log_{\mathbf{x}}^K(\mathbf{y}) = \sqrt{K}\operatorname{arcosh}\left(-\langle\frac{\mathbf{x}}{\sqrt{K}},\frac{\mathbf{y}}{\sqrt{K}}\rangle_{\mathcal{L}}\right)\frac{\mathbf{y} + \frac{1}{K}\langle\mathbf{x},\mathbf{y}\rangle_{\mathcal{L}}\mathbf{x}}{||\mathbf{y} + \frac{1}{K}\langle\mathbf{x},\mathbf{y}\rangle_{\mathcal{L}}\mathbf{x}||_{\mathcal{L}}}.
$$

$\square$

## B.2 Curvature

**Lemma 1.** *For any hyperbolic spaces with constant curvatures $-1/K, -1/K' > 0$, and any pair of hyperbolic points $(\mathbf{u},\mathbf{v})$ embedded in $\mathbb{H}^{d,K}$, there exists a mapping $\phi : \mathbb{H}^{d,K} \to \mathbb{H}^{d,K'}$ to another pair of corresponding hyperbolic points in $\mathbb{H}^{d,K'}$, $(\phi(\mathbf{u}),\phi(\mathbf{v}))$ such that the Minkowski inner product is scaled by a constant factor.*

*Proof.* For any hyperbolic embedding $\mathbf{x} = (x_0, x_1, \ldots, x_d) \in \mathbb{H}^{d,K}$ we have the identity: $\langle\mathbf{x},\mathbf{x}\rangle_{\mathcal{L}} = -x_0^2 + \sum_{i=1}^{d} x_i^2 = -K$. For any hyperbolic curvature $-1/K < 0$, consider the mapping $\phi(\mathbf{x}) = \sqrt{\frac{K'}{K}}\mathbf{x}$. Then we have the identity $\langle\phi(\mathbf{x}),\phi(\mathbf{x})\rangle_{\mathcal{L}} = -K'$ and therefore $\phi(\mathbf{x}) \in \mathbb{H}^{d,K'}$. For any pair $(\mathbf{u},\mathbf{v})$, $\langle\phi(\mathbf{u}),\phi(\mathbf{v})\rangle_{\mathcal{L}} = \frac{K'}{K}\left(-\mathbf{u}_0\mathbf{v}_0 + \sum_{i=1}^{d}\mathbf{u}_i\mathbf{v}_i\right) = \frac{K'}{K}\langle\mathbf{u},\mathbf{v}\rangle_{\mathcal{L}}$. The factor $\frac{K'}{K}$ only depends on curvature, but not the specific embeddings. $\square$

Lemma 1 implies that given a set of embeddings learned in hyperbolic space $\mathbb{H}^{d,K}$, we can find embeddings in another hyperbolic space with different curvature, $\mathbb{H}^{d,K'}$, such that the Minkowski inner products for all pairs of embeddings are scaled by the same factor $\frac{K'}{K}$.

For link prediction tasks, Theorem 4.1 shows that with infinite precision, the expressive power of hyperbolic spaces with varying curvatures is the same.

**Theorem 4.1.** *For any hyperbolic curvatures $-1/K, -1/K' < 0$, for any node embeddings $H = \{\mathbf{h}_i\} \subset \mathbb{H}^{d,K}$ of a graph $G$, we can find $H' \subset \mathbb{H}^{d,K'}$, $H' = \{\mathbf{h}_i'|\mathbf{h}_i' = \sqrt{\frac{K'}{K}}\mathbf{h}_i\}$, such that the reconstructed graph from $H'$ via the Fermi-Dirac decoder is the same as the reconstructed graph from $H$, with different decoder parameters $(r,t)$ and $(r',t')$.*

*Proof.* The Fermi-Dirac decoder predicts that there exists a link between node $i$ and $j$ iif $\left[e^{(d_{\mathcal{L}}^K(\mathbf{h}_i,\mathbf{h}_j)-r)/t} + 1\right]^{-1} \geq b$, where $b \in (0,1)$ is the threshold for determining existence of links. The criterion is equivalent to $d_{\mathcal{L}}^K(\mathbf{h}_i,\mathbf{h}_j) \leq r + t\log(\frac{1-b}{b})$.

Given $H = \{\mathbf{h_1}, \ldots, \mathbf{h_n}\}$, the graph $G_H$ reconstructed with the Fermi-Dirac decoder has the edge set $E_H = \{(i,j)|d_{\mathcal{L}}^K(\mathbf{h}_i,\mathbf{h}_j) \leq r + t\log(\frac{1-b}{b})\}$. Consider the mapping to $\mathbb{H}^{d,K'}$, $\phi(\mathbf{x}) := \sqrt{\frac{K'}{K}}\mathbf{x}$. Let $H' = \{\phi(\mathbf{h_1}), \ldots, \phi(\mathbf{h_n})\}$. By Lemma 1,

$$
d_{\mathcal{L}}^{K'}(\phi(\mathbf{h}_i),\phi(\mathbf{h}_j)) = \sqrt{K'}\operatorname{arcosh}\left(-\frac{K'}{K}\langle\mathbf{h}_i,\mathbf{h}_j\rangle_{\mathcal{L}}/K'\right) = \sqrt{\frac{K'}{K}}d_{\mathcal{L}}^K(\mathbf{h}_i,\mathbf{h}_j). \qquad (20)
$$

| Name | Nodes | Edges | Classes | Node features |
|---|---|---|---|---|
| CORA | 2708 | 5429 | 7 | 1433 |
| PUBMED | 19717 | 88651 | 3 | 500 |
| HUMAN PPI | 17598 | 5429 | 4 | 17 |
| AIRPORT | 3188 | 18631 | 4 | 4 |
| DISEASE | 1044 | 1043 | 2 | 1000 |
| DISEASE-M | 43193 | 43102 | 2 | 1000 |

Table 3: Benchmarks' statistics

Due to linearity, we can find decoder parameter, $r'$ and $t'$ that satisfy $r' + t' \log(\frac{1-b}{b}) = \sqrt{\frac{K'}{K}}(r + t \log(\frac{1-b}{b}))$. With such $r'$, $t'$, the criterion $d_{\mathcal{L}}^{K}(\mathbf{h}_i, \mathbf{h}_j) \leq r + t \log(\frac{1-b}{b})$ is equivalent to $d_{\mathcal{L}}^{K'}(\phi(\mathbf{h}_i), \phi(\mathbf{h}_j)) \leq r' + t' \log(\frac{1-b}{b})$. Therefore, the reconstructed graph $G_{H'}$ based on the set of embeddings $H'$ is identical to $G_H$. $\qquad\square$

## C   Experimental Details

### C.1   Dataset statistics

We detail the dataset statistics in Table 3.

### C.2   Training details

Here we present details of HGCN's training pipeline, with optimization and incorporation of DropConnect [42].

**Parameter optimization**. Recall that linear transformations and attention are defined on the tangent space of points. Therefore the linear layer and attention parameters are Euclidean. For bias, there are two options: one can either define parameters in hyperbolic space, and use hyperbolic addition operation [10], or define parameters in Euclidean space, and use Euclidean addition after transforming the points into the tangent space. Through experiments we find that Euclidean optimization is much more stable, and gives slightly better test performance compared to Riemannian optimization, if we define parameters such as bias in hyperbolic space. Hence different from shallow hyperbolic embeddings, although our model and embeddings are hyperbolic, the learnable graph convolution parameters can be optimized via Euclidean optimization (Adam Optimizer [19]), thanks to exponential and logarithmic maps. Note that to train shallow Poincaré embeddings, we use Riemannian Stochastic Gradient Descent [4, 48], since its model parameters are hyperbolic. We use early stopping based on validation set performance with a patience of 100 epochs.

**Drop connection**. Since rescaling vectors in hyperbolic space requires exponential and logarithmic maps, and is conceptually not tied to the inverse dropout rate in terms of re-normalizing L1 norm, Dropout cannot be directly applied in HGCN. However, as a result of using Euclidean parameters in HGCN, DropConnect [42], the generalization of Dropout, can be used as a regularization. DropConnect randomly zeros out the neural network connections, *i.e.* elements of the Euclidean parameters during training time, improving the generalization of HGCN.

**Projections**. Finally, we apply projections similar to Equations 16 and 17 for the hyperboloid model $\mathbb{H}^{d,K}$ after each feature transform and $\log$ or $\exp$ map, to constrain embeddings and tangent vectors to remain on the manifold and tangent spaces.