[Reviews · NeurIPS 2019]

Reviewer 1



The paper is well written in general although it contains mistakes and ignores some related work. In particular, it is not clear whether the corollaries (whose proofs are given in the appendix) are sold as contributions or not. Many of their implications are already known in the machine learning literature (see details below). == Mistakes: - Wrong curvature (minor mistake): The hyperboloid defined in Eq. (3) is said to have a constant curvature of -1/K^2 in the submission. As explained in detail in Section 3.4 of Chapter 3 of the second edition of [1A] (or also in the following references [1C] and [1D]), its curvature is actually -1/K. The explanation in these references uses the fact that the hyperboloid can be seen as a half spherical model with imaginary radius equal to i * sqrt(K). From this duality between the spherical and hyperbolic models, many formulations (e.g. exponential and logarithm maps, parallel transport etc...) can be infered from one model to the other (see details for the mistake below). - Wrong parallel transport formulation (major mistake): The parallel transport formula in Eq. (15) (supp. material) is wrong. The denominator of the right term should be the squared geodesic distance instead of the geodesic distance. The details can for example be found in the last equation of Section B.1 of [1B], which is given for the spherical case although the hyperbolic case is very similar due to the duality between the spherical and hyperbolic models. This can be verified by observing that the Minkowski inner product between examples belonging to tangent space (e.g. of the north pole in H^{d,1} in the paper) should be preserved via parallel transport. I provide a code in Matlab below to show that the provided parallel transport equation is wrong. If the parallel transport formula in Eq. (15) was correct, then r1 would be equal to r3 (obtained by using Eq. (15)). However, r1 and r2 (obtained by using the hyperbolic adaptation of [1B]) are equal. I did not manage to check if the source code also exploited that formulation of the parallel transport. == Connections to existing works It is not clear if the different theorems in Section 3 (Background section) are sold as contributions since the proofs are given. These results either correspond to definitions or are already known in the literature. For instance, Corollary 3.1 contains definitions and seems to be useful to obtain a Poincare distance that depends on the curvature. The version of the Poincare distance that depends on the curvature (Eq. (4) of the submission) is already known in the machine learning literature. It is for example given in Eq. (18) of [1C]. The hyperbolic exponential and logarithm maps that depend on the curvature are also given in Section 6.1 of [1C]. Theorem 4.1 is very similar to the last paragraph of the proof given in Section A.2 of [1D]. Law et. al [1D] also observe that decreasing the curvature of the hyperbolic space improves classification and retrieval performances. Nonetheless, they do not exploit multiple layers of hyperbolic representations. == Experiments and baselines The experimental results (node classification (with F1 score) or unseen link predictions between nodes (with ROC evaluation metric)) show huge improvement of the hyperbolic GCN when each layer has a specific curvature. The comparison to Euclidean baselines is not really fair since their architecture contains many more hyperparameters which are: - the hyperboloid radii (one for each layer of the GCN) - the bias r in Eq. (14) - the temperature t in Eq. (14) Each curvature can be seen as a hyperparameter, this curvature is equal to 0 in the Euclidean case. The paper mentions in line 225 that having different curvatures is very important for good performance. Fortunately, Table 2 shows an ablation study with less tuning of the hyperparameters. In conclusion, the main contributions of the submission then seem to be only the adaptation of GCNs to the hyperbolic case, and huge experimental improvement compared to Euclidean baselines. minor comments: - the MLP acronym is first used in line 186 although it is defined in line 272. - there is already a model named HyperGCN (Yadati et al.) [1A] Peter Petersen, Riemannian Geometry, 2006 [1B] Bergmann, R., Fitschen, J. H., Persch, J., & Steidl, G. , Priors with coupled first and second order differences for manifold-valued image processing, Journal of mathematical imaging and vision, 60(9) [1C] Wilson, R., Hancock, E., Pekalska, E., Duin, R., Spherical and Hyperbolic Embeddings of Data, IEEE T-PAMI, 2014 [1D] Law, M., Liao, R., Snell, J., Zemel, R., Lorentzian Distance Learning for Hyperbolic Representations, ICML 2019 (rejected by ICLR 2019, so publicly available since September 2018) [1E] Ganea O., Becigneul G., Hofmann T., Hyperbolic Neural Networks, NIPS 2018 == code showing that Eq. (15) is not correct (using a curvature of -1 for simplicity) d = 4; o = 10*rand(d-1,1); o = [sqrt(sum(o.^2) + 1);o]; %o = [1;0;0;0]; y = 10*rand(d-1,1); y = [sqrt(sum(y.^2) + 1);y]; a = create_tangent_vector(o); b = create_tangent_vector(o); r1 = minkowski_inner_product(a,b) r2 = minkowski_inner_product(parallel_transport(a, o, y), parallel_transport(b, o, y)) r3 = minkowski_inner_product(parallel_transport_submission(a, o, y), parallel_transport_submission(b, o, y)) function a = create_tangent_vector(o) a = 10*rand(length(o)-1,1); a0 = (a'*o(2:end)) / o(1); a = [a0;a]; end function r = minkowski_inner_product(x,y) e = x .* y; e(1) = -e(1); r = sum(e); end function r = hyperboloid_normalization(x) abs_norm = sqrt(minkowski_inner_product(x,x)); r = x / abs_norm; end function r = geodesic_distance(x,y) r = acosh(-minkowski_inner_product(x,y)); end function r = logarithm_map(x,y) r = y + minkowski_inner_product(y,x) * x; r = hyperboloid_normalization(r) * geodesic_distance(x,y); end function r = parallel_transport(xi, x, y, used_power) if nargin < 4 used_power = 2; end r = xi - minkowski_inner_product(logarithm_map(x,y), xi) / (geodesic_distance(x,y)^used_power) * (logarithm_map(x,y) + logarithm_map(y,x)); end function r = parallel_transport_submission(xi, x, y) r = parallel_transport(xi, x, y, 1); end edit after rebutall: The authors confirmed that the parallel transport formula they used was correct, so I am okay with it. The theoretical novelty is limited but in terms of experimental contributions, the paper is interesting.

Reviewer 2



General Comments: The paper is very well written although very dense. I enjoyed the supplementary material, it helped me cover some background knowledge that I needed to understand the paper. The material is very original and very useful since it improves the original goal of hyperbolic neural networks (model graph and tree-like data). The best way is to actually combine it with Graph NNs that we already know perform really well. The experiments are extensive and convincing line 27:” due to the fact that the number of elements/nodes increases exponentially with hierarchy, but the area in Euclidean space grows linearly with distance.” Which area exactly? The area of a disk grows quadratically with respect to the radious not linearly. Also, this argument is fundamentaly wrong as it compares an infinite space with a finite space. The number of the leafs of a hierarchical space grows exponentially but remains finite. The area of a disk contains infinite points. Technically speaking the unit circle can fit the universe. The reason why euclidean space is different and it is described very well on this paper: As described in this paper https://papers.nips.cc/paper/5971-space-time-local-embeddings.pdf The maximum number of points which can share a common nearest neighbor is limited 2 for 1-dimensional spaces, 5 for 2-dimensional spaces while such centralized structures do exist in real data d-dimensional spaces can at most embed (d + 1) points with uniform pair-wise similarities. See the references: K. Zeger and A. Gersho. How many points in Euclidean space can have a common nearest neighbor? In International Symposium on Information Theory, page 109, 1994. L. van der Maaten and G. E. Hinton. Visualizing non-metric similarities in multiple maps. Machine Learning, 87(1):33–55, 2012. I suggest reading this blog https://networkscience.wordpress.com/2011/08/09/dating-sites-and-the-split-complex-numbers/ and also this paper: https://dl.acm.org/citation.cfm?id=2365942 line 91: “In contrast to previous work, we derive core neural network operations in a more stable model” That statement is not backed. It is a bit in the air. You have to be more specific. One of the weaknesses of this paper is that it doesn’t stress the differences with Reference [10] Hyperbolic Neural Networks. Especially on the derived operators. From my understanding, the contribution is on aggregation layers and attention mechanisms. It also seems to me that another difference is that you work on the Lorentz manifold (which you call hyperboloid, it would be better to use Lorentz, since it is more widely known as that in the ML community).

Reviewer 3



A novel hyperbolic graph convolutional network is proposed in this submission, which shows outstanding performance on the graphs possessing hierarchical structure in the link prediction and link prediction tasks. The only concern the reviewer has is the definition of activation function defined in equation 10. It means that after implementing a normal activation function on T_o^{k_{l-1}}, the intermediate vector defined in this tangent space is directly mapped to a space with radius k_l via exp_o^{k_l}, is this mapping even mathematically plausible?

[Author Response · NeurIPS 2019]

We really appreciate the reviewers' time and effort. Thank you for your detailed feedback, thoughtful suggestions and valuable recommendations for improving the paper! The reviewers appreciate our strong empirical results and mainly asked us to better situate our work. We respond in more detail below, and will take all comments into account in our revised version. Additionally, we will shortly release open-source PyTorch implementations of the Hyperbolic Graph Convolution Networks (HGCN) model and baselines, along with our detailed reproducible training setup.

**Contributions (R1, R2, R3):** We sincerely thank R1 for their careful read and for pointing out ambiguities in our paper. First, our paper is an empirical paper. Our goal is to develop practical techniques that can improve the predictive performance of recently-developed Graph Convolutional Networks (GCNs) using ideas from hyperbolic geometry. Our main result is that HGCN achieves error reduction of up to 63.1% in ROC AUC for link prediction and of up to 47.5% in F1 score for node classification. Moreover, using standard notions from hyperbolic geometry (Gromov's $\delta$-hyperbolicity), we show that performance is indeed improved when the underlying graph is more "hyperbolic-like".

Simply setting GCN variables to be optimized in hyperbolic space does not yield good performance. Our work validates that three algorithmic ideas based on hyperbolic geometry are important to obtaining predictive accuracy and good runtime performance: trainable curvature, attention-based hyperbolic aggregation, and optimization on the Lorentz (hyperboloid) manifold. Our aggregation method relies on two crucial techniques which result in improved performance compared to standard aggregation in the tangent space at the origin: (1) aggregation is performed at the *local* tangent space of each point, which better approximates the local hyperbolic geometry; (2) attention scores are computed from the origin, which allows HGCN to capture node hierarchies. Furthermore, trainable curvature intuitively helps find the right amount of curvature, and potentially alleviates numerical errors that might arise from limited machine precision. We show in ablation analysis that these algorithmic contributions result in up to 9.9% absolute gain compared to simple GCN in hyperbolic space, an improvement that is larger than what any Euclidean GCN variant achieves.

**Presentation (R1, R2, R3):** The results in Section 3 are not considered part of our contribution and we now realize that we could make our claims and presentation more crisp. In our updated draft, we will state known facts from hyperbolic geometry as propositions rather than corollaries, move standard results to the Appendix as suggested by R2, and more carefully cite the related literature, including [1], as suggested by R1. We have also fixed the typo in Equation 15 in the Appendix pointed out by R1 and we confirm that we have been using the correct formulation of parallel transport in our experiments. Indeed, we had run unit tests verifying that points are mapped to the correct tangent space. We also thank R1 for noticing the notation error regarding the hyperbolic radius $i\sqrt{K}$, and we will make this consistent in the revised version.

We thank R2 for the thorough feedback and will clarify our intuitive explanation on the hyperbolic volume growth property in line 27, which was meant to illustrate the fact that the volume of balls in Euclidean space grows polynomially with respect to the radius, while in hyperbolic space it grows exponentially. We will also replace the hyperboloid manifold by the Lorentz manifold in the revised version.

R3 also asks about the correctness of Equation 10, which maps tangent spaces with different curvatures. In order to apply the exponential map at the north pole, we need to make sure that points are located in the corresponding tangent space. Fortunately, tangent spaces of the north pole are shared across hyperboloid manifolds that have different curvatures, making equation 10 mathematically correct as long as $\sigma(0) = 0$, which is true for the ReLU activation. We will make this more explicit in the revised version.

**Experiments (R1, R2):** R1 rightly points out that the number of hyper-parameters should be the same for fair comparison between models. In our experiments, we were very careful about this and we ensured fairness of all methods by controlling the number of hyper-parameters and trainable parameters. We will clarify this detail in the revision. In particular, curvatures in HGCN are trainable parameters which are therefore not subject to hyper-parameter search. Regarding bias and temperature hyper-parameters, we consistently used the Fermi-Dirac decoder to predict link probabilities for our method and all baselines (hyperbolic and Euclidean), observing that it performs the same as the standard dot product decoder commonly used in Euclidean baselines.

R2 requests clarification regarding the stability of the hyperboloid model compared to the Poincaré model. In our experiments, we used both models observing that they achieve similar performance, but that the hyperboloid model offers more stable optimization. This statement is also supported by the fact that the Poincaré distance function is numerically unstable due to the denominator term, confirming a previous observation in [2]. We will clarify this point in the revision.

**Connection to prior work (R2):** We appreciate R2's suggestion to discuss the connections to Hyperbolic Neural Networks (HNN). The main differences are the use of trainable curvatures, the attention-based hyperbolic aggregation mechanism, and the use of hyperboloid model. In the revised manuscript, we will further highlight and stress these key components and add an experiment using HNN in the hyperboloid model, as suggested by R2.

[1] M. Law, R. Liao, J. Snell, and R. Zemel. "Lorentzian Distance Learning for Hyperbolic Representations." ICML 2019.
[2] M. Nickel, and D. Kiela. "Learning Continuous Hierarchies in the Lorentz Model of Hyperbolic Geometry." ICML 2018.


[Meta-Review · NeurIPS 2019]

This paper develops a Graph Convolutional Network that works in hyperbolic space. The development is a relatively straightforward analog of other neural network models that have been adapted to hyperbolic space, but all reviewers agree the experimental results are interesting. There were some mistakes in the original submission and a lack of clarity about whether the theoretical results were being claimed as novel. The authors have clarified that the key mistake was a typo and the correct setting was used in the experiments, which satisfied R1. The authors also clarified that they are not claiming novelty of the theoretical results. This needs to be made more clear in the camera ready. Having said this, all reviewers agree the experimental contributions are interesting. R2 and R3 supported acceptance, and R1 said they were ok with that outcome.